# Unveiling the Lipid Features and Valorization Potential of Atlantic Salmon (*Salmo salar*) Heads

**DOI:** 10.3390/md22110518

**Published:** 2024-11-15

**Authors:** João Pedro Monteiro, Tiago Sousa, Tânia Melo, Carla Pires, António Marques, Maria Leonor Nunes, Ricardo Calado, M. Rosário Domingues

**Affiliations:** 1Centro de Espetrometria de Massa & LAQV-REQUIMTE & Departamento de Química, Universidade de Aveiro, Campus Universitário de Santiago, 3810-193 Aveiro, Portugal; tmms@ua.pt (T.S.); taniamelo@ua.pt (T.M.); 2CESAM & Departamento de Química, Universidade de Aveiro, Campus Universitário de Santiago, 3810-193 Aveiro, Portugal; 3CIVG—Vasco da Gama Research Center/EUVG—Vasco da Gama University School, 3020-210 Coimbra, Portugal; 4Division of Aquaculture, Upgrading and Bioprospection, Portuguese Institute for the Sea and Atmosphere (IPMA, I.P.), Av. Doutor Alfredo Magalhães Ramalho 6, 1495-165 Algés, Portugal; cpires@ipma.pt (C.P.); amarques@ipma.pt (A.M.); 5Interdisciplinary Centre of Marine and Environmental Research (CIIMAR/CIMAR-LA), University of Porto, Terminal de Cruzeiros do Porto de Leixões, Av. General Norton de Matos s/n, 4450-208 Matosinhos, Portugal; nunes.leonor@gmail.com; 6ECOMARE & CESAM & Departamento de Biologia, Universidade de Aveiro, Campus Universitário de Santiago, 3810-193 Aveiro, Portugal

**Keywords:** aquaculture, omega-3 fatty acids, fish co-products, lipidomics, nutritional quality, salmon, sustainability

## Abstract

The sustainable utilization of co-products derived from the salmon processing industry is crucial for enhancing the viability and decreasing the environmental footprint of both capture and aquaculture operations. Salmon (*Salmo salar*) is one of the most consumed fish worldwide and a major species produced in aquaculture. As such, significant quantities of salmon co-products are produced in pre-commercialization processing/steaking procedures. The present study characterized a specific co-product derived from the processing of salmon: minced salmon heads. More specifically, this work aimed to reveal the nutritional profile of this co-product, with a special focus on its lipid content, including thoroughly profiling fatty acids and fully appraising the composition in complex lipids (polar lipids and triglycerides) for the first time. The antioxidant potential of lipid extracts from this salmon co-product was also studied in order to bioprospect lipid functional properties and possibly unveil new pathways for added-value applications. Our analysis indicated that these minced salmon heads are exceptionally rich in lipids. Oleic acid is the most prevalent fatty acid in this co-product, followed by palmitic acid, stearic acid, and linoleic acid. Moreover, relevant lipid indexes inferred from the fatty acid composition of this co-product revealed good nutritional traits. Lipidome analysis revealed that triglycerides were clearly the predominant lipid class present in this co-product while phospholipids, as well as ceramides, were also present, although in minimal quantities. The bioprospecting of antioxidant activity in the lipid extracts of the minced salmon heads revealed limited results. Given the high concentration of triglycerides, minced salmon heads can constitute a valuable resource for industrial applications from the production of fish oil to biodiesel (as triglycerides can be easily converted into fatty acid methyl esters), as well as possible ingredients for cosmetics, capitalizing on their alluring emollient properties. Overall, the valorization of minced salmon heads, major co-products derived from the processing of one of the most intensively farmed fish in the world, not only offers economic benefits but also contributes to the sustainability of the salmon processing industry by reducing waste and promoting a more efficient use of marine bioresources.

## 1. Introduction

Atlantic salmon (*Salmo salar*) is a critical resource for human nutrition, normally recognized for its high-quality protein content and for the presence of beneficial fatty acids (FAs), namely omega-3 [1,2,3,4]. As one of the most consumed fish globally, its production is both indispensable and substantial. In 2023, global salmon production reached approximately 3 million metric tons [5], with aquaculture accounting for about 70% of this volume [6]. This high level of production generates significant quantities of co-products including heads, bones, skin, and viscera. Estimates suggest that the fillet yield could represent only 50–70% of the whole salmonid weight [7], implying a production of roughly up to 1.5 million metric tons of salmon co-products annually.

A considerable number of studies have studied the lipid content of *S. salar* muscle/edible tissue, including characterizations at a molecular level and the use of lipidomic approaches [8,9,10,11,12,13,14], which is proof of the interest and relevance of this species. Regarding salmon co-products in particular, the phospholipid composition of the head, roe, and skin (although in a different salmon species, *Oncorhynchus tshawytscha*) was studied using nuclear magnetic resonance spectroscopy [15]. This study showed that salmon heads contained less phospholipid and omega-3 fatty acid content (including eicosapentaenoic (EPA) and docosahexaenoic (DHA) acids) than roe and skin [15]. The other available report using lipidomics to study salmon co-products characterized the phospholipid extracts of the heads of an unspecified salmon species using more standard liquid chromatography coupled with a mass spectrometry (LC-MS) approach [16]. This study reported high percentages of DHA and EPA in the fatty acid chains of phosphatidylcholine (PC) and phosphatidylethanolamine (PE), the two main phospholipid classes present [16]. Other studies using lipidomics tools to characterize the lipid content of the heads of other fish species included those conducted on the silver carp (*Hypophthalmichthys molitrix*) [16], Pacific blue mackerel (*Scomber australasicus*) [17], and king salmon (*Oncorhynchus tshawytscha*) [15], in the later species using a nuclear magnetic resonance (NMR), with all focusing specifically on the content in omega-3-containing phospholipids. This interest in the content in omega-3-containing phospholipids is justified by the wide range of health benefits ascribed to these compounds when acquired through the diet [18,19]. Other studies simply focused on the characterization of the fatty acid profiles of *S. salar* co-products in specific, unveiling desirable features in terms of the omega-3 fatty content in heads [20,21], as well as in frames, skin trimmings, and viscera [21]. Despite these reportedly enticing characteristics of the lipid content of salmon co-products, the bioprospecting of bioactive lipids in salmon co-products with a perspective of valorizing these resources for higher-end applications in the industry still represents an unexplored path.

The exploitation of salmon co-products as prospective sources of valuable compounds was the subject of numerous studies, mostly focused specifically on the protein content and related bioprospecting of the biological activity of protein fractions and hydrolysates [22,23,24,25,26,27,28,29,30,31,32,33,34]. Despite the lipid content of salmon co-products remaining rather less explored in terms of the bioprospecting of biological activity and the isolation of bioactive lipids, a few preliminary assessments assigned antioxidant [35], anti-inflammatory [35,36], antithrombotic [37], cardioprotective [37], and antimicrobial activity [35,38] to lipid fractions and oils obtained from salmon heads. Also, orally administered phospholipid extracts from salmon heads were linked to ameliorating effects upon rodent models used to study a metabolic syndrome [16]. Therefore, further bioprospecting and characterizing, in detail, the lipid fractions of salmon co-products could substantiate the pertinence of recovering bioactive lipids or lipid fractions from these bioresources, envisioning biorefinery approaches that allow taking full advantage of both protein and lipid contents [39].

The valorization of fish co-products is paramount for the sustainability and economic viability of the seafood industry [39]. Using these co-products can significantly reduce waste, lower the environmental impact, and create new revenue streams [40,41]. Making use of state-of-the-art technical resources including gas chromatography–mass spectrometry (GC-MS) and liquid chromatography–mass spectrometry (LC-MS), this study aimed to achieve an in-depth characterization of the whole lipid content of Atlantic salmon heads. Moreover, since the available studies characterizing the lipid content in salmon co-products at a molecular level only focused on phospholipids, we aimed to extend the characterization of minced salmon heads to their whole lipidome. The bioprospecting of biological activity in the lipid extracts of minced salmon heads could also pave the way for novel, higher-end applications for these bioresources in the industry. We believe that this type of characterization of fish co-products is instrumental to unveiling new applications and/or directing these resources to the most suitable industrial uses, contributing to a more sustainable and economically viable farmed fish industry framed by circularity and contributing more efficiently to a blue bioeconomy.

## 2. Results

### 2.1. Elemental Composition and Biochemical Characterization

The results of the elemental analysis quantifying the contents of carbon, hydrogen, nitrogen, and sulfur are depicted in Appendix A. In terms of the biochemical characterization of the minced salmon heads, the most striking result was the remarkable content of lipids (23.97 ± 0.72% WW), which was much higher than the content of protein (16.20 ± 20% WW; Table 1). This salmon co-product also displayed an abundant moisture content (58.04 ± 0.99% WW) and very limited contents of ash (0.81 ± 0.22% WW) and carbohydrates and other compounds (0.97 ± 0.80% WW; Table 1), which were determined as the remaining component percentages. Another remarkable feature of this co-product was the considerably low phospholipid content in the total lipid fractions obtained from these resources (1.16 ± 0.17% of total lipid; Table 1).

### 2.2. Fatty Acid Profiles

The fatty acid profile of the minced salmon heads was characterized by a very significant content of fatty acids with 18-carbon chains. Oleic acid (C18:1*n*-9) was clearly the most abundant fatty acid present (34.66 ± 1.26%, Figure 1; 116.0 ± 15.4 μg/mg DW, Appendix A) while stearic (C18:0) and linoleic acids (C18:2*n*-6) were also among the major fatty acids present (11.64 ± 1.14% and 12.10 ± 0.41%, respectively; Figure 1). Palmitic acid (C16:0) was the other fatty acid that was present in more noticeable amounts, representing 13.24 ± 1.44% of all the fatty acid content (Figure 1). Omega-3 fatty acids were present at modest amounts and included linolenic acid (C18:3*n*-3; 3.70 ± 0.12%, Figure 1; 15.8 ± 2.2 μg/mg DW, Appendix A), eicosapentaenoic acid (EPA, C20:5*n*-3, 2.70 ± 0.12%, Figure 1; 12.6 ± 1.7 μg/mg DW, Appendix A), and docosahexaenoic acid (DHA, C22:6*n*-3, 2.86 ± 0.11%, Figure 1; 12.7 ± 1.4 μg/mg DW, Appendix A). In Appendix A, the results of the fatty acid profiling are presented in absolute quantities.

Monounsaturated fatty acids (MUFAs) represented almost half of the total fatty acid content (46.59 ± 1.98%; Table 2) while the contents of saturated (SFA) and polyunsaturated fatty acids (PUFA) were lower (28.29 ± 2.56% and 25.12 ± 0.79%, respectively). The minced salmon heads contained more omega-6 (or *n*-6, 12.44 ± 0.38%) than omega-3 (or *n*-3, 10.68 ± 0.35%) fatty acids, although the resulting *n*-6/*n*-3 ratio was still reasonably low (1.17 ± 0.02), as was the saturated-to-unsaturated (SFA/UFA) ratio (0.39 ± 0.05; Table 2). The average fatty acid chain length was very close to 18 (18.01 ± 0.04), confirming the significant contributions of fatty acids like oleic (the most abundant), stearic, and linoleic acids to the total content of fatty acid. This salmon co-product was also shown to present interestingly low atherogenic (AI) and thrombogenic (TI) indexes (0.28 ± 0.03 and 0.43 ± 0.06, respectively), along with a remarkably high hypocholesterolemic/hypercholesterolemic (h/H) ratio (0.42 ± 0.05; Table 2).

### 2.3. Lipidome Characterization

Analysis of the lipidome of the minced salmon heads by LC-MS allowed the identification and semi-quantification of 150 different lipid species (Figure 2). This analysis revealed that triglycerides (TGs) were not only the most abundant but also the most diverse class of lipids present (Figure 2; Appendix A). In fact, 108 different TG species were identified, including one oxidized TG form (TG 50:2;O). Other acylglycerides, namely diglycerides (DGs) were also present, with 14 different species being detected, all in low amounts. In terms of the polar lipid content and phospholipids in particular, the detected species were also present in residual amounts, as was already somewhat inferred from the quantification of phospholipids in lipid total extracts (Table 1). Nevertheless, fifteen PC species and five lysophosphatidylcholines (LPCs) were detected. Other phospholipid classes, namely molecular PEs (2), lysophosphatidylethanolamine (LPE, 1) phosphatidylglycerol (PG, 1) and phosphatidylserine (PS, 2), were barely represented (Figure 2; Appendix A). Two sphingolipids, one ceramide (Cer 42:2;2O), and one hexosylceramide (HexCer 42:2;2O) were also detected.

In terms of the specific profiles of TGs, the most prevalent lipid class present in samples of minced salmon heads (Appendix A), 108 different molecular species were recorded, with total fatty acyl carbon chains ranging from 37 to 66 total carbons and with the most unsaturated species being TG 62:13 (Figure 3), corresponding to a TG specifically containing one oleic and two DHA esterified fatty acids. Further analysis of the TG profile revealed that TG 54:3 (containing three esterified oleic acids) was the most abundant TG species, representing 16.20 ± 0.35% of the content of the class (Figure 3 and Figure 4). The only other two TG species with contents representing over 10% of the class were TG 52:2 (with the major form containing one palmitic and two oleic esterified fatty acids, representing 11.54 ± 0.32% of total TGs) and TG 52:3 (with the major form containing one palmitic, one oleic, and one linoleic esterified fatty acid, amounting to 10.04 ± 0.15% of the class; Figure 4).

The percentage of marine TGs (as defined as the ones containing EPA and/or DHA) was also ascertained, representing 12.71 ± 0.22% of the total lipid content of the class.

### 2.4. Screening of Antioxidant Activity

The antioxidant activity of total lipid extracts of minced salmon heads was assessed by the use of both DPPH^●^ and ABTS^●+^ assays (Table 3). Lipid extracts from minced salmon heads were shown to display a limited free radical scavenging capacity. In fact, it was only possible to determine 10.76% of the inhibition for the lipid extracts of minced salmon heads in the DPPH^●^ assays and 15.41% in the ABTS^●+^ assay using 250 μg mL^−1^ of the lipid extract (Table 3). The DPPH^●^ assay also resulted in low values in terms of TE for the minced salmon head lipid extract, with a value of 4.39 for the DPPH^●^ assay and a value of 1.67 in the ABTS^●+^ assay.

## 3. Discussion

The proximate composition of the minced salmon heads showed a remarkable amount of lipid content, certainly one of the highest reported for fish and other seafood co-products [39]. Atlantic salmon is considered a high fat fish species, according to tentative classifications [42], and the lipid content of their heads seems to corroborate that assumption. Thus, salmon heads represent an obvious possibility for exploring this naturally abundant content in lipids for novel applications, for which a great first step would always be a thorough characterization of lipid composition at molecular level.

The results presented here for the proximate composition, in general, agree with results reported in previous studies. Vásquez et al. [23] reported values of 62.6% of moisture (WW) and contents of 54.3% of lipids and 30.1% of protein (DW), as well as a very restricted ash content (2.7%), in *S. salar* heads. Another study by Malcorps and colleagues reported the moisture content to range between 50 and 55% and confirmed the remarkable quantity of lipids (>40% DW), which was higher than the protein content (≈30%), describing, however, a significantly higher ash content in salmon heads (≈8%) [21]. It is important to mention that reports studying the proximate composition of salmon flesh/fillets highlight very meaningful compositional features regarding salmon heads, underscoring nutritional quality [1,3,4] and nutraceutical potential [20,36,37]. These reports systematically propose that in salmon flesh (unlike in the case of minced salmon heads), the composition of protein is higher than that of lipids (sometimes three times larger in relative terms) [1,3,4]. Other studies directly comparing the fat (lipid) content in salmon heads to that in salmon flesh/fillets corroborated that the first displayed a content much higher in terms of lipids than proteins [20,36]. Another remarkable feature of salmon heads was the notably low content of total phospholipids present in total lipid extracts. This overwhelming predominance of neutral lipids and TGs in particular (accounting for well over 90% of lipids) in salmon heads had already been highlighted in previous studies [20,37].

Regarding the fatty acid profile of salmon head homogenate, our study showed fatty acids with 18-carbon chains to be especially enriched in this co-product, especially oleic acid, representing more than a third of total fatty acid content. Other studies characterizing the fatty acid profile of Atlantic salmon heads were in line with our findings, namely confirming oleic acid to consistently be the main fatty acid present, and highlighted linoleic and palmitic acids as the main components of fatty acid profiles [20,21,35,36,38]. However, these studies also generally reported lower stearic acid (C18:0) levels than the ones reported in the present study (2.3–5.4%) [20,21,35,36,38]. When considering the contents of marine omega-3 fatty acids in salmon heads, the literature reports vary significantly, with some pointing to quantities of EPA similar to the ones reported in the present work but higher contents of DHA [21,36]. Nonetheless, other studies referred higher relative quantities of both EPA and DHA [20,35] while other works completely failed to detect/report EPA or DHA [38]. Methodological differences, both in lipid extraction methods and in the derivatization and determination of fatty acid content, may account at least for some of the discrepancies between these reports. Moreover, growth conditions, especially the diet provided to fish during grow-out, may also contribute to the disparities reported in salmon head fatty acid profiles [43,44,45]. Also, the fatty acid profile of commercially available *S. salar* was reported to vary significantly according to several production specifications, namely ranges (value, standard, premium, or organic) and origins [3], making it particularly challenging to compare the fatty acid profiles of their heads without any background information on diet and other grow-out conditions (e.g., water temperature). A study characterizing the fatty acid profiles of Atlantic salmon fillets from ten different retailers clearly documented this variability, with oleic acid, despite being consistently the most abundant fatty acid (as in *S. salar* heads), varying between 24.3 and 42.0% [3] while other reports described an even lower content (17.7% in wild *S. salar*) [1]. Linoleic acid ranged from 8.3 to 15.1% and palmitic acid from 8.7 to 14.1% as examples of other main fatty acid components [3]. The contents of marine omega-3 FAs were found to be very variable, with the EPA ranging from 2.6 to 8.5% and DHA from 3.1 to 9.4% [1,3,4].

Omega-3 fatty acids collectively, and EPA and DHA in particular, have been ascribed a wide range of health benefits [46,47,48,49,50,51], with a combined intake of EPA + DHA from 250 to 500 mg/day in adults being recommended by health organizations [52]. Our calculations showed that less than 50 g of salmon head homogenate (DW) would be enough to meet those requirements. Therefore, even though the content in salmon heads of omega-3 fatty acids, namely EPA and DHA, is somewhat restricted in relative terms when compared to that of other fatty acids, the richness of salmon heads in lipid contents makes them good prospective sources of these fatty acids in qualitative or absolute terms. Moreover, despite this aforementioned restricted content of EPA and DHA, minced salmon heads still present a very enticing *n*-6/*n*-3 ratio (of 1.17). An *n*-6/*n*-3 ratio lower than 4–5 is recommended in the diet [53,54] as presenting several health benefits [55,56] and actively mitigating the prejudicial effects of the high *n*-6/*n*-3 ratios characteristic of Western diets [57,58]. Moreover, the saturated-to-polyunsaturated (SFA/PUFA) fatty acid index, one of the other most commonly used indexes for appraising the nutritional value of dietary foods, is also in line with that described for the edible parts of many other fish species, or even generally lower [59]. Additionally, minced salmon heads also present low thrombogenic indexes (TIs) and remarkably high hypocholesterolemic/hypercholesterolemic (h/H) indexes when generally compared to other fish species [59], which represent additional beneficial features in nutritional terms.

The analysis of the lipidome of the minced Atlantic salmon heads allowed detecting TGs containing small-chain fatty acids (8–12 carbons), as well as other odd-chain fatty acids in their fragmentation profiles (MS/MS) that were not detected by GC-MS. This fact is explained by a lower sensitivity of the GC-MS technique with regard to high resolution LC-MS, and by the fact that the TG species containing those fatty acids were vestigial with regard to the majority TG species. This study also highlighted the predominance of TGs as the most well-represented lipid class in this co-product. A study also characterizing the phospholipid content in salmon heads also reported residual contents of phospholipids, such as phosphatidylinositol (PI) and sphingomyelin (SM), in salmon heads, in addition to the phospholipid classes that were detected in our study [20]. Another study using lipidomics focused specifically on the characterization of phospholipid extracts from salmon heads [16], detecting molecular species of the following lipid classes: LPA, LPC, LPE, LPG, LPI, LPS, PA, PC, PE, PG, PI, and PS [16]. However, in our case, we were not able to detect lipids from the LPA, LPG, LPI, LPS, PA, and PG and PI classes given the overwhelming predominance of TGs in our total lipid extracts [16]. This previous study reported PC 34:1 as being the most abundant PC in salmon heads and PE O-34:2 to be the most abundant molecular species from the PE class [16]. In our study, PC 34:1 was, in fact, one of the most abundant PCs in the minced salmon heads, along with PC 42:6 and PC 38:6, while we were not able to detect ether lipids belonging to the PE lipid class. In the same previous study, the percentage of phospholipids containing EPA and DHA was larger than 20%, therefore being higher than the percentage of TGs containing those fatty acids reported here [16]. This enrichment of the polar lipid components of Atlantic salmon heads with regard to the results presented here for TGs was confirmed in another study, reporting EPA and DHA contents in polar lipid extracts to attain 10.1 and 16.8%, respectively [37]. Moreover, in king salmon (*Oncorhynchus tshawytscha*), this trend was also recorded, with polar lipid fractions of salmon heads being reported to be especially enriched in EPA and DHA with regard to the neutral lipid content [15].

A study specifically characterizing the TG profile in salmon muscle tissue through lipidomics identified TG 58:7 (16.4% of the class total) as the major TG present in the lipid extracts of these co-products, followed by TG 54:5 (11.6%) [11]. These results mean that the TG composition in the salmon muscle differs considerably from that of the head. Indeed, the most abundant TG in the head is TG 54:3 (16.2%) while the main TG in the muscle (TG 58:7) only occurs residually (≈0.02%) in the head. Two other different studies also employing a lipidomic approach to characterize the fillets/muscle of Atlantic salmon and focusing only on its phospholipid content identified PC 38:6 and PE 38:6 as the most abundant molecular species for each phospholipid class [8,10]. In salmon heads, PC 38:6 was among the most abundant PC present, along with PC 42:6, while we were not able to detect PE 38:6.

With regard to works characterizing the heads of other fish species by lipidomics tools, the available studies are difficult to compare to ours because they focused specifically on the content in omega-3-containing phospholipids. A study characterizing silver carp heads phospholipids detected lipid species from the LPC, LPE, LPG, LPI, LPS, phosphatidic acid (PA), PC, PE, PG, PI, and PS classes [16]. Here, we were not able to detect lipids from the LPG, LPI, LPS, PA, and PI classes, probably because of the fact that the polar lipid content is almost negligible in total lipid extracts of minced salmon heads. With regard to the classes that we were able to detect, this study reported LPC 16:0, LPE 22:6, PC 34:1, PE-O 34:2, PG 34:1, and PS 36:1 to be the most abundant lipid species of their class [16]. In our study, although we were only able to detect a restricted number of phospholipid species, we were able to detect LPC 16:0 and PC 34:1, and LPE 22:6 was, in fact, the only member of its class that we detected.

While the analysis of the lipid content and profile of salmon heads revealed a somewhat limited content in omega-3 fatty acids, and the marine fatty acids EPA and DHA, when compared to other animal seafood co-products [39], that content, in absolute terms, ended up being very considerable given the substantial content of lipids present in this salmon co-product. The global demand for omega-3 fatty acids was estimated to attain 3.61 billion by 2028 [60]. This projection intensified demand and consolidated the nutritional importance of omega-3 fatty acids, mostly due to the plethora of beneficial health-promoting benefits generally ascribed to the consumption of these fatty acids (namely EPA and DHA). These biomolecules have been reported to yield beneficial effects in the mitigation of neurodegenerative and cardiovascular diseases and cancer and protection against inflammation specifically [61,62,63,64,65,66,67,68,69]. However, the production of fish oils containing marine omega-3 fatty acids is under pressure, both from the side of demand, because of current demographics, and from the side of production, due to climate change impacts and overfishing [70,71]. Therefore, the search for new alternative sources of omega-3 fatty acids (namely EPA and DHA) is now more intensive than ever [72]. Fish co-products have already been profusely explored for the production of fish oils containing the marine omega-3 fatty acids EPA and DHA [73,74,75,76,77,78]. Given the volume of production and consumption of salmon at a worldwide scale, along with the remarkable level of lipid content in salmon heads, the situation can make these resources very enticing for food/feed applications. In this sense, given the current demand for omega-3 fatty acids, lipid fractions obtained from salmon heads could be potentially more readily explored [79,80,81]. Moreover, the similarities between salmon heads and olive oil, namely the high content of oleic acid [82,83,84], also overwhelmingly predominant in the form of triglycerides [85,86,87], may also be promising for future applications given the range of health benefits already reported for these biomolecules, especially in the context of cardiovascular disease and metabolic syndrome [88,89,90,91,92]. It is important to notice, however, that eventual uses of these resources for food/feed should take into account that lipids, particularly those containing PUFA, may be readily degraded by lipid oxidation reactions with the production of secondary oxidation products contributing to flavor deterioration and the occurrence of off-flavors [93,94,95]. Moreover, some of these products may present toxicity risks [96,97,98]. Therefore, the development of new approaches to detect and quantify oxidized lipids in resources like these co-products is important, and studies like this, characterizing the lipid components at a molecular level, represent a first step in that direction.

In the cosmetics industry, TGs from salmon heads may also represent valuable resources taking advantage of the emollient qualities of TGs [99]. In this case, this salmon co-product may represent an alternative not only to vegetable fat but also to other animal fat sources that have already been exploited in the industry [100,101,102,103]. Rendered poultry fat, tallow, and lard are widely used in the cosmeceutical industry for their occlusive, emulsion-stabilizing, emollient, surfactant, and viscosity-increasing properties [104]. The natural origin of these lipids can be a significant marketing point, catering to the increasing consumer demand for natural and sustainably sourced cosmetic ingredients. However, the purposeful use of salmon head lipids for cosmeceutical applications is still constrained by a general lack of bioprospecting of biological activity and bioactive lipids in the lipid fractions of these co-products.

This salmon co-product has, therefore, valorization potential in the food/feed and cosmetics industry. But another opportunity for its valorization while those applications are implemented and developed comes from an increasingly challenged energy industry. In fact, another alternative suitable application could include the production of biodiesel [105,106] and the production of skin care products. In biodiesel production, TGs represent primary materials due to their high energy content and efficient conversion to FAMEs [107,108,109]. The production of biodiesel from waste animal fat is rather established and presents great potential as these resources do not compete with the final production of food products and also contribute to a global reduction in waste produced by the food industry [106,110]. Reports have shown that the use of animal fat for the production of biodiesel has approximately doubled in the past decade [111]. Moreover, the demand for animal fat for biofuel production is projected to triple by 2030 with regard to the beginning of this decade [111]. The use of salmon-head-derived TGs may represent a valid alternative to vegetable oils and other animal fat sources [112,113,114] for biodiesel production, enabling an alternative and renewable energy source that can reduce reliance on fossil fuels and lower greenhouse gas emissions. The importance of salmon for human nutrition and the production volume of this fish species [115] make salmon co-products readily available bioresources. Interestingly, oleic acid in particular, by far the most abundant fatty acid in salmon heads, is a major component in almost all biodiesel formulations and was reported to present highly favorable features for biodiesel production, namely a convenient balance between oxidative stability and low-temperature operability [116,117,118,119,120]. Moreover, the very low levels of phospholipids in the lipid extracts of salmon heads may represent an additional advantage from an industrial point of view, simplifying the processing and refinement of these resources to highly concentrated TG fractions.

Here, we reported the lipid extracts of salmon heads to display moderate antioxidant activity. The lipid content in salmon heads and other salmon co-products remains, for the most part, rather unexplored in terms of the bioprospecting of biological activities and bioactive lipids, with the exception of a handful of studies. Salmon backbones, heads, and viscera oils obtained by Soxhlet and microwave-assisted extractions were previously tested for cytotoxic, antioxidant, anti-inflammatory, and antimicrobial activities with promising results [35]. As also reported here, the antioxidant activity of salmon head lipids was rather limited, and even only detected when salmon head oil was obtained specifically by one of the techniques employed (microwave-assisted extraction) [35]. The anti-inflammatory potential of lipid extracts from the heads of *S. salar* was confirmed in another study [36], as were the antimicrobial properties of oils produced from *S. salar* co-products: in this case, oils from the heads, but also discarded soft tissues [38]. Another study reported lipid extracts heads of *S. salar* to display antithrombotic activity and potential cardioprotective properties [37]. Lastly, phospholipid extracts from salmon heads were shown to display general ameliorating effects in rodent models of metabolic syndrome [16]. However, despite the promise of these bioprospecting studies and the remarkable lipid content of salmon heads, much remains to be explored in terms of screening for biological activity and bioactive lipids in salmon heads in particular and salmon co-products in general.

Other than lipids, the protein content of salmon co-products has also been explored for novel added-value applications. Not only the heads of salmon but also the trimmings, backbones/frames, viscera, tailfins, and the skin were all explored for the recovery of valuable protein fractions, gelatin, collagen, and protein hydrolysates [22,23,24,25,26,27,30,31,32,33,34]. Moreover, protein extracts and hydrolysates from salmon co-products (including the heads) have been reported to yield multiple bioactive properties [22,23,24,26,27,28,29,30]. These promising bioactivities and applications targeting the protein content of salmon co-products, including heads, along with the rather untapped potential of their lipids, may sustain and justify an integrated biorefinery approach that optimally recovers both proteins and lipids and directs these biomolecules for selected high-end uses. Such an approach would not only increase the profitability of the salmon industry by creating multiple new revenue streams but also contribute to its sustainability by minimizing waste and promoting a circular economy. In this study, the priority was to characterize, as deeply and thoroughly as possible, the lipid content in these resources to fully unveil their potential in terms of their lipid content. For this purpose, we used the traditional extraction methods (the Bligh and Dyer method, more specifically [121]) known to guarantee the highest yield and coverage of lipid classes. It is obvious that if food/feed applications are to be prioritized, lipid extraction, either alone or in biorefinery setups, must be explored using green extraction methods and solvents. In this case, our study offers a good baseline for comparison and for an eventual optimization of other methods.

The limitations of this work include the fact that the lipidomics approach involved a semi-quantitative procedure, and therefore, it is not possible to calculate, quantitatively, the amounts of each lipid species and lipid classes per quantity of biomass of the co-product. Moreover, since the objective of this work was to perform a characterization of the lipid content as thoroughly as possible, we used the standard methods (Bligh and Dyer) known to assure an optimized extraction of total lipid and lipid class coverage. However, these classical methods use solvents that are not compatible with uses in the food/feed industry and, therefore, the next steps of the valorization of these co-products include testing the use of green methods/solvents for the lipid extraction in these resources. Another limitation was the fact that we were only able to bioprospect biological activity in lipid extracts from these resources in terms of antioxidant activity. In the future, it will be important to extend this screening to other possible manifestations of biological activity in order to improve the possibilities of unveiling other innovative uses for these resources in possible higher-end applications. Finally, future work also includes the planning and testing of biorefinery setups able to optimally recover and guarantee the quality of both lipid and protein contents from these co-products.

## 4. Materials and Methods

### 4.1. Chemicals

High-grade liquid chromatography (HPLC)-grade dichloromethane (CH_2_Cl_2_), 96% absolute ethanol (CH_3_CH_2_OH), and methanol (CH_3_OH) were obtained from Fisher Scientific Ltd. (Loughborough, UK). Milli-Q purified water was obtained using the Synergy^®^ system from Millipore Corporation (Billerica, MA, USA). Whatman No. 1 filter paper was acquired from Sigma-Aldrich, St. Louis, MO, USA. The 2,2′-azino-bis(3-ethylbenzothiazoline-6-sulfonic acid) radical cation (ABTS^●+^) was obtained from Fluka (Buchs, Switzerland) while the α,α-diphenyl-β-picrylhydrazyl radical (DPPH^●^) was obtained from Sigma-Aldrich (St. Louis, MO, USA). The 37 Component FAME Mix was purchased from Supelco (Sigma-Aldrich, St. Louis, MO, USA) along with the internal standard methyl nonadecanoate (≥99% purity) from Sigma-Aldrich (St. Louis, MO, USA). Lipid internal standards for lipidomic analyses were acquired from Avanti Polar Lipids, Inc. (Alabaster, AL, USA), including 1,2-dimyristoyl-*sn*-glycero-3-phosphate (dMPA), 1,2-dimyristoyl-*sn*-glycero-3-phospho-(1′-rac-glycerol) (dMPG), 1,2-dimyristoyl-*sn*-glycero-3-phosphocholine (dMPC), 1,2-dipalmitoyl-*sn*-glycero-3-phosphatidylinositol (dPPI), 1,2-dimyristoyl-sn-glycero-3-phosphoethanolamine (dMPE), 1,2-dimyristoyl-*sn*-glycero-3-phosphatidylserine (dMPS), 1-nonadecanoyl-2-hydroxy-*sn*-glycero-3-phosphocholine (LPC), N-heptadecanoyl-D-erythro-sphingosine (Cer), 1′,3′-bis[1,2-di-tetradecanoyl-*sn*-glycero-3-phospho]-*sn*-glycerol (CL), and N-heptadecanoyl-D-erythro-sphingosylphosphorylcholine (SM). All other reagents were purchased from leading commercial suppliers guaranteeing product quality.

### 4.2. Samples

Fresh heads of farmed Atlantic salmon (*Salmo salar*) were purchased at a local supermarket and transported to the laboratory on ice. Heads were ground and minced in an industrial meat grinder (HOBART, Troy, OH, USA) and stored at −20 °C until analysis. Heads were mechanically ground and minced in an industrial meat grinder (HOBART, Troy, OH, USA) and stored at −20 °C until analysis. These co-products did not include any additive since only mechanical means are used in the industrial processing. Five different portions of minced heads were subsequently subjected to all the analyses described in this section.

### 4.3. Biochemical and Elemental Composition

Moisture and ash contents were assessed using standard analytical methods. Five portions of minced salmon heads (250 mg) were placed in ceramic crucibles and dried overnight in an oven set at 105 °C. Following a 24 h period, the crucibles were cooled to room temperature in a desiccator containing silica gel and weighed to determine moisture content. Ash determination followed a two-step procedure. First, the salmon-head mince was pre-incinerated by placing the crucibles on a heated plate for 20 min. Subsequently, the crucibles were transferred to a muffle furnace and maintained at 575 °C for 6 h. Once cooled, the crucibles were weighed to determine the ash content.

The elemental composition of lyophilized co-product samples, specifically carbon (C), hydrogen (H), nitrogen (N), and sulfur (S) contents, was analyzed using a Leco Truspec-Micro CHNS 630-200-200 elemental analyzer. The combustion furnace temperature was set at 1075 °C, with the afterburner set to 850 °C. Approximately 2 mg of minced salmon heads was combusted in an oxygen/carrier gas mixture, ensuring complete combustion and conversion of co-products to water vapor, carbon dioxide, and nitrogen for gas analysis. Carbon, hydrogen, and sulfur were detected via infrared absorption while nitrogen was measured using thermal conductivity. This elemental analysis enabled the estimation of protein content using a nitrogen-to-protein conversion factor. The standard conversion factor of 6.25 was employed as it had commonly been used in previous studies to determine protein content in salmon [3,20,21,23]. Lipid content was measured by gravimetry after lipid extraction, and carbohydrates (with other compounds) represented the remaining wet weight percentage after accounting for moisture, ash, lipid, and protein contents.

### 4.4. Lipid Extraction

Total lipid extracts from salmon head homogenate were obtained using the Bligh and Dyer method, with slight adaptations [121,122]. Approximately 10 mg of freeze-dried co-product samples were thoroughly minced and homogenized with a mortar and pestle and placed in glass tubes. To these tubes, 2.5 mL of methanol and 1.25 mL of dichloromethane were added. The mixture was vortexed for 1 min and another 1.25 mL was added to each sample-containing tube. After vortexing for another minute, the mixtures were incubated for 30 min in an orbital shaker on ice (4 °C) and then centrifuged at 626× *g* for 10 min at room temperature. The supernatant was transferred to a new glass tube and the precipitate (biomass) was re-extracted two times (adding 2.5 mL methanol and 1.25 mL dichloromethane each time, followed by 1 min vortexing and centrifugation at 626× *g* for 10 min at room temperature, with subsequent collection of the supernatant into the same tube). The combined organic phases were completely dried under a nitrogen stream, re-dissolved in 2 mL of dichloromethane and 2 mL of methanol, and vortexed for 1 min. Ultra-pure water (1.8 mL) was added to each tube; the mixtures were vortexed for 1 min and centrifuged at 626× *g* for 10 min at room temperature. Organic (lower) phases were collected into new tubes and the aqueous (upper) phases were re-extracted with 2 mL of dichloromethane, followed by 2 min of vortexing and a final centrifugation at 626× *g* for 10 min at room temperature. The lower organic phases were combined in the same tube, filtered (Whatman No. 1 filter paper), dried under a nitrogen stream, and preserved at −20 °C for further analysis. Finally, the total lipid content in salmon heads was estimated by gravimetric analysis.

### 4.5. Phospholipid Quantification in Total Lipid Extracts

The phospholipid quantification in total lipid extracts of minced salmon heads was performed using a modified version of Bartlett and Lewis method [122,123,124]. Briefly, samples were dissolved in 300 μL of dichloromethane, and small aliquots (10 μL, in duplicates) were transferred to acid-washed glass tubes and evaporated under a nitrogen stream. Each sample-containing tube was then added 125 μL of perchloric acid (HClO_4_, 70% m/V) and incubated for 60 min at 180 °C in a steel heating block. Following the hydrolysis step, 850 μL of water, 125 μL of ammonium molybdate (2.5%, m/V), and 125 μL of ascorbic acid (10%, m/V) were added to the content of each tube. The mixture was then vortexed and incubated for 10 min at 100 °C in a water bath. A calibration curve was prepared using standards with known phosphorus concentrations ranging from 0.1 to 2.0 μg (using a standard solution of NaH_2_PO_4_·2H_2_O containing 100 μg of phosphorus per mL). Absorbance measurements were performed at 797 nm using a Thermo Scientific Multiskan Go microplate UV-vis spectrophotometer (Thermo Scientific, Hudson, NH, USA).

### 4.6. Gas Chromatography–Mass Spectrometry (GC-MS)

The fatty acid content in lipid extracts of minced salmon heads was analyzed by GC-MS following transmethylation. Thirty microgram aliquots of the lipid extracts were transferred to glass tubes and dried under a nitrogen stream. The lipid films were then dissolved in 1 mL of *n*-hexane containing C19:0 as an internal standard (1 μg mL^−1^, CAS number 1731-94-8, Merck, Darmstadt, Germany). Each tube was added 200 μL of a potassium hydroxide (KOH) 2 M solution in methanol and the mixture was vortexed for 2 min. Next, 2 mL of a saturated sodium chloride (NaCl) solution was added and the mixture was centrifuged for 5 min at 626× *g* to promote phase separation. Cholesterol in the upper (organic) phase was removed using a protocol available on the Lipid Web “https://lipidhome.co.uk/ms/basics/msmeprep/index.htm (accessed on 5 May 2024). A 10 mm silica column in a pipette tip with wool was pre-conditioned with 5 mL of hexane. Methyl esters were added to the top of the column and eluted with a hexane ether (95:5, *v*/*v*, 3 mL) mixture, then completely dried under a nitrogen stream. Finally, fatty acid methyl esters (FAMEs) were dissolved in 100 μL of *n*-hexane, and 2 μL of the resulting solution was injected into an Agilent Technologies 8860 GC System (Santa Clara, CA, USA) equipped with a DB-FFAP column (30 m length, 0.32 mm internal diameter, and 0.25 μm film thickness, J&W Scientific, Folsom, CA, USA). The gas chromatograph was connected to an Agilent 5977B Network Mass Selective Detector, operating with electron impact ionization at 70 eV, scanning the mass range of *m*/*z* 50–550 in a 1 s cycle in full scan mode. The oven temperature program started at 58 °C for 2 min and increased by 25 °C min^−1^ to 160 °C, by 2 °C min^−1^ to 210 °C, and by 30 °C min^−1^ to 250 °C, holding for 10 min. The injector and detector temperatures were set at 220 °C and 280 °C, respectively. Helium was used as the carrier gas at a flow rate of 1.4 mL min^−1^. Fatty acids were identified by comparing retention times to those of commercial FAME standards in the Supelco 37 Component FAME Mix (ref. 47885-U, Sigma-Aldrich, Darmstadt, Germany) and by MS-spectrum comparison with chemical databases (Wiley 275 library and AOCS lipid library). The relative percentages of fatty acids were calculated using the percent relative area method and quantification was performed by normalization with the included internal standard methyl nonadecanoate (19:0). Various indexes, including average chain length (ACL), double bond index (DBI), peroxidizability index (PI), atherogenic index (AI), thrombogenic index (TI), hypocholesterolemic/hypercholesterolemic index (h/H), and polienic index (PoI), were calculated as previously described [122,125].

### 4.7. Reverse-Phase Liquid Chromatography–Mass Spectrometry (C18–LC–MS)

Total lipid extracts from minced salmon heads were analyzed by reverse-phase liquid chromatography (C18-LC-MS) using a Dionex Ultimate 3000 (Thermo Fisher Scientific, Bremen, Germany) with an Ascentis^®^ Express 90 Å C18 column (Sigma-Aldrich^®^, 2.1 × 150 mm, 2.7 µm) coupled to a Q-Exactive^®^ hybrid quadrupole Orbitrap mass spectrometer (Thermo Fisher, Scientific, Bremen, Germany). The mobile phases used for the gradient during RP-LC-MS determinations consisted of mobile phase A (Milli-Q water/acetonitrile (40/60%) with 10 mM ammonium formate (NH_4_HCO_2_) and 0.1% formic acid (CH_2_O_2_)) and mobile phase B (isopropanol/acetonitrile (90/10%) with 10 mM ammonium formate and 0.1% formic acid). The gradient defined for the experiments was as follows: 32% B at 0 min, 45% B at 1.5 min, 52% B at 4 min, 58% B at 5 min, 66% B at 8 min, 70% B at 11 min, 85% B at 14 min, 97% B at 18 min, 97% B at 25 min, 32% B at 25.01 min, and 32% B at 33 min. A mixture containing 1 µg of lipid extract from salmon head homogenate was prepared in 91 µL of a solvent system consisting of 50% isopropanol/50% methanol and 8 µL of a mixture of phospholipid standards (dMPC—0.04 µg, SM d18:1/17:0—0.04 µg, dMPE—0.04 µg, LPC—0.04 µg, dPPI—0.08 µg, CL(14:0)4—0.16 µg; dMPG—0.024 µg, Cer 17:0/d18:1—0.08 μg, dMPS—0.08 µg, and dMPA—0.16 µg) and loaded into the C18 column at 50 °C with a flow rate of 260 µL min^−1^. The mass spectrometer operated simultaneously in positive (3.0 kV) and negative (−2.7 kV) modes. The capillary temperature was set to 320 °C and the sheath gas flow to 35 U. Data acquisition was performed in full-scan mode with a high resolution of 70,000 and automatic gain control (AGC) target of 3 × 10^6^, in an *m*/*z* range of 300–1600, with 2 micro scans and a maximum injection time (IT) of 100 ms. Tandem mass spectra (MS/MS) resolution was 17,500, with an AGC target of 1 × 10^5^, 1 micro scan, and a maximum IT of 100 ms. The cycles included a full-scan mass spectrum and 10 data-dependent MS/MS scans, continuously repeated throughout the experiments with a dynamic exclusion of 30 s and an intensity threshold of 8 × 10^4^. The normalized collision energy (CE) ranged between 20, 24, and 28 eV in the negative mode and 25 and 30 eV in the positive mode. Data acquisition was performed using the Xcalibur data system (V3.3, Thermo Fisher Scientific, Bremen, Germany). Molecular lipid species were identified using the Lipostar software version 2.1.5 (Molecular Discovery Ltd., Borehamwood, UK) [126]. This software is capable of processing raw-data importing, peak detection, integration, and identification. Lipid assignment and identification were performed against a database created from the LIPID MAPS structure database (version of June 2024). The database was fragmented using the DB Manager Module of Lipostar, following Lipostar fragmentation rules. The raw files were directly imported and aligned according to the settings defined by Lange et al. [127]. Automatic peak picking was carried out with the SDA smoothing level set to high and a minimum signal-to-noise ratio of 3. Automatic isotope clustering settings included a tolerance of 7 ppm and a retention time tolerance of 0.2 min. The MS/MS filter was applied to retain features with MS/MS spectra for identification. Lipid identification was performed using the following parameters: 5 ppm precursor ion mass tolerance and 10 ppm product ion mass tolerance. Lipostar annotations and assignments were manually confirmed based on the analysis of the fragmentation (MS/MS spectra) and the presence of the *m*/*z* signaling the presence of the fatty acids and class-characteristic fragments. The areas of the peaks of each lipid species were normalized by calculating the ratio against the area of the respective class internal lipid standard included at a known concentration. The relative abundance of each lipid species was estimated by dividing the normalized peak areas of each lipid species by the sum of the total normalized peak areas.

### 4.8. Antioxidant Activity

The antioxidant scavenging activities of minced salmon heads total lipid extracts against α,α-diphenyl-β-picrylhydrazyl (DPPH^●^) and 2,2′-azino-bis-3-ethylbenzothiazoline-6-sulfonic acid radical cation (ABTS^●+^) were evaluated using previously described methods [125,128]. Briefly, 150 μL of an ethanolic dilution of the lipid extracts (50, 250, and 500 μg mL^−1^) was combined with 150 μL of DPPH^●^ or ABTS^●+^ working solution in ethanol (absorbance ≈ 0.9). The samples were incubated for 120 min and the absorbance was measured at 517 nm for DPPH^●^ and 734 nm for ABTS^●+^ every 5 min using a UV-vis spectrophotometer (Multiskan GO 1.00.38, Thermo Scientific, Hudson, NH, USA). Controls were prepared by substituting the radical solution with ethanol. To ensure the stability of the radicals, solutions with the radical plus ethanol were also prepared. All measurements were performed in triplicate. The same procedure was applied to the Trolox standard solution (12.5, 62.5, 125, 250 μg mL^−1^ in ethanol). The antioxidant activity, as expressed as the percentage of inhibition of the DPPH^●^ (or ABTS^●+^), was calculated using Equation (1), which is as follows:Inhibition (%) = ((Abs_Radical_ − (Abs_Sample_ − Abs_Control_))/Abs_Radical_) × 100(1)
Here, Abs_Radical_ is the absorbance of the radical (DPPH^●^ or ABTS^●+^), Abs_Sample_ is the absorbance of the sample with radical (DPPH^●^ or ABTS^●+^), and Abs_Control_ is the absorbance of the sample with ethanol.

The antioxidant activity expressed in Trolox equivalents (TE) was calculated according to Equation (2):TE (µmol g^−1^) = IC Trolox (µmol g^−1^) × 1000/IC of samples (µg mL^−1^)(2)
Here, IC is the concentration of lipid extract per sample and of Trolox, which promotes inhibition at the tested concentrations of the extract in the radicals DPPH^●^ or ABTS^●+^.

### 4.9. Statistical Analysis

Figures were produced using GraphPad Prism version 7.00 for Windows (GraphPad Software, La Jolla, CA, USA). All experimental data are shown as means ± standard deviations (SDs) for 5 samples of minced salmon heads (*n* = 5).

## 5. Conclusions

Given its significant role in human nutrition and its importance as one of the most produced and preferred fish species for human consumption worldwide, the salmon industry will continue to generate considerable quantities of co-products that cannot be regarded as waste. Therefore, the efficient use of salmon co-products is paramount to foster sustainability in fisheries and aquaculture. One of the most important findings of this study was the remarkably rich lipid content of minced salmon heads (23.97 ± 0.72 WW). Such a substantial lipid content should warrant further study, particularly targeting bioprospecting and valorization pathways. The lipid content of minced salmon heads occurs predominantly in the form of triglycerides (TGs), paving the way for multiple uses (Figure 5). A biorefinery setup, which optimizes the extraction of both lipids and proteins contents, could represent a promising approach to handle these resources. This dual extraction strategy could lead to a more comprehensive utilization of salmon heads, contributing to the economic and environmental sustainability of the salmon industry. This study proved that the biochemical characterization of co-products, in particular the elucidation of the lipid content, can play a key role in identifying novel uses for these bioresources, which, in the case of these particular resources, may include food/feed and cosmeceutical applications. Overall, the present study ultimately highlighted the importance of a thorough characterization of fish co-products for the optimization of their use and value, ultimately supporting a more sustainable and economically viable fish industry under a blue bioeconomy framework.

## Figures and Tables

**Figure 1 marinedrugs-22-00518-f001:**
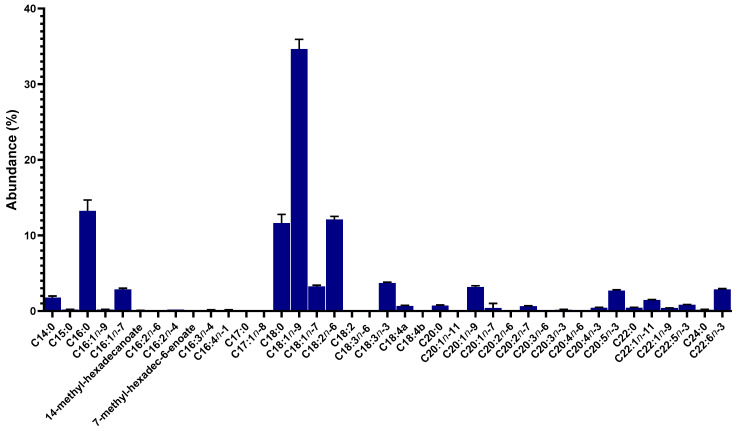
Fatty acid profile of the minced salmon heads (results are presented as percentages of total fatty acid content). Data are shown as means ± standard deviations (SDs) for 5 samples of salmon head homogenate (*n* = 5).

**Figure 2 marinedrugs-22-00518-f002:**
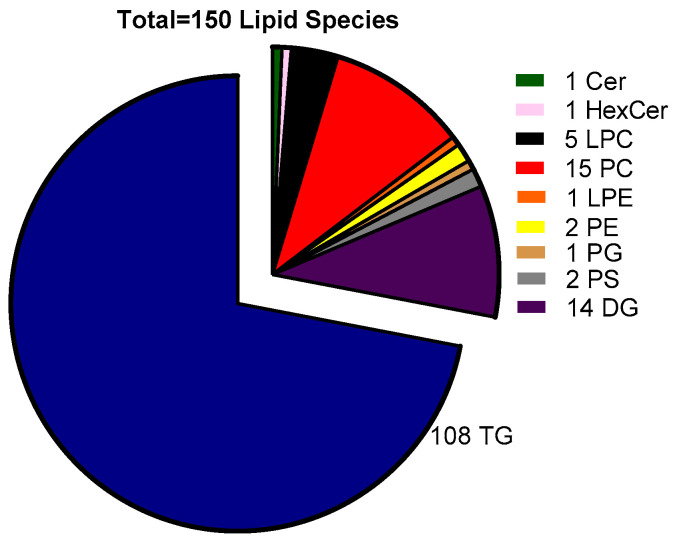
Number of molecular lipid species detected in minced salmon heads according to lipid class.

**Figure 3 marinedrugs-22-00518-f003:**
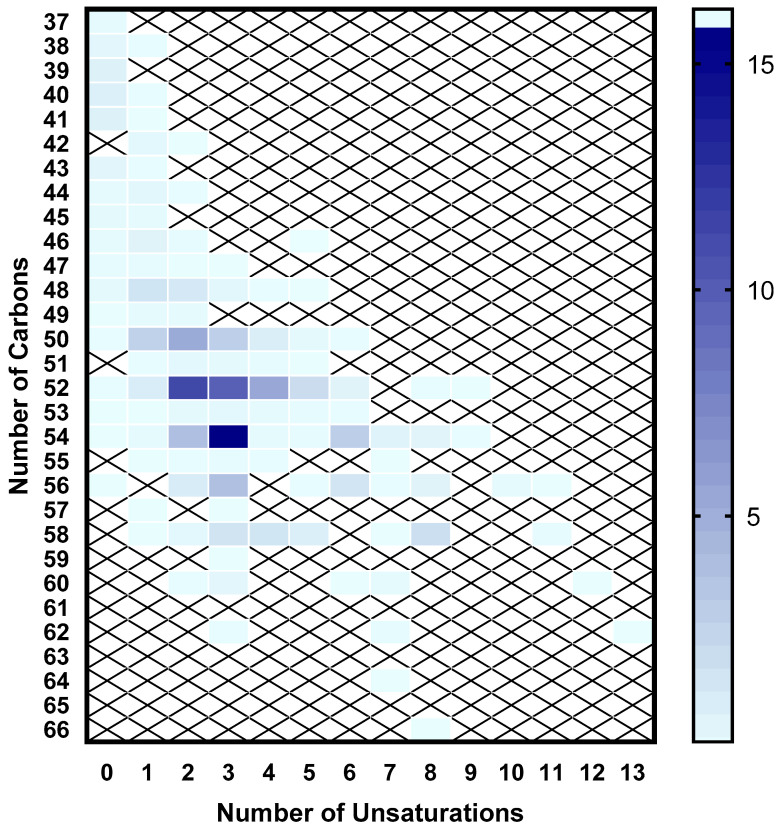
Heatmap depicting the percentages of triglycerides present in minced salmon heads as distributed according to total acyl chain length and unsaturation degree.

**Figure 4 marinedrugs-22-00518-f004:**
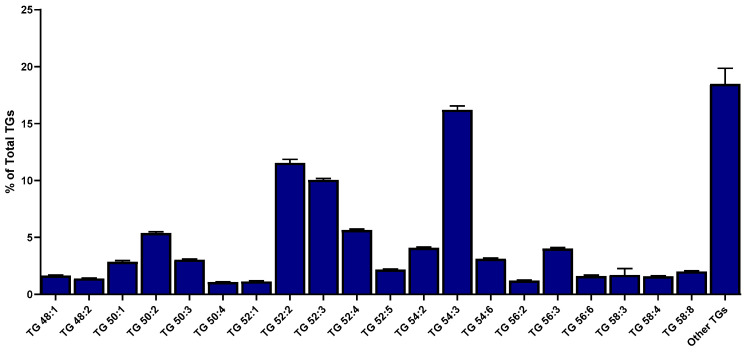
Percentages of molecular lipid species of triglycerides present in minced salmon heads in terms of the total content of the class (only the top 20 most abundant species are presented).

**Figure 5 marinedrugs-22-00518-f005:**
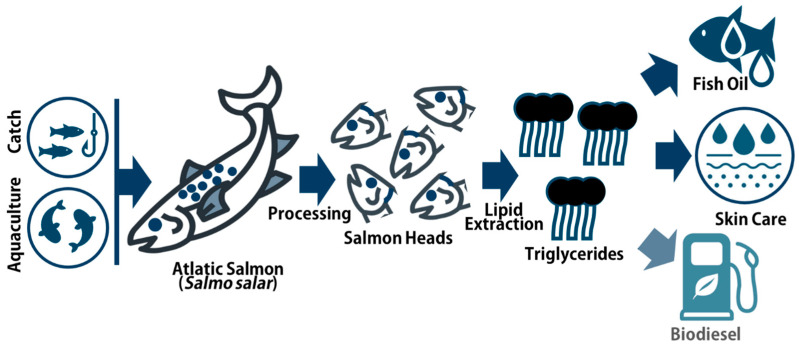
Salmon heads as source of concentrated triglyceride fractions and potential industry applications as pathways or valorization.

**Table 1 marinedrugs-22-00518-t001:** Proximate composition of the minced salmon heads. Data are shown as means ± standard deviations (SDs) for 5 samples of salmon head homogenate (*n* = 5).

Component (WW)	%
Moisture	58.04 ± 0.99
Ash	0.81 ± 0.22
Protein	16.20 ± 1.02
Carbohydrates and other compounds	0.97 ± 0.80
Lipid	23.97 ± 0.72
Phospholipid ^1^	1.16 ± 0.17

^1^ Phospholipid content is presented as percentage of total lipid content.

**Table 2 marinedrugs-22-00518-t002:** Indexes/factors derived from the fatty acid profiles of minced salmon heads. Data are shown as means ± standard deviations (SDs) for 5 samples of salmon head homogenate (*n* = 5).

Index/Factor	Value
*n*-3	10.68 ± 0.35%
*n*-6	12.44 ± 0.38%
*n*-6/*n*-3	1.17 ± 0.02
SFA	28.29 ± 2.56%
MUFA	46.59 ± 1.98%
PUFA	25.12 ± 0.79%
SFA/PUFA	1.13 ± 0.12
TI	0.43 ± 0.06
AI	0.28 ± 0.03
h/H	4.24 ± 0.49
PoI	0.42 ± 0.05
UI	125.59 ± 3.93
ACL	18.01 ± 0.04

ACL: average chain length; AI: atherogenic index; h/H: hypocholesterolemic/hypercholesterolemic index; MUFA: monounsaturated fatty acid; PoI: polienic index; PUFA: polyunsaturated fatty acid; SFA: saturated fatty acid; UI: unsaturation index; TI: thrombogenic index.

**Table 3 marinedrugs-22-00518-t003:** Free radical scavenging capacity of the total lipid extracts of minced salmon heads expressed as inhibition percentages for 250 μg mL^−1^ of lipid extract for DPPH^●^ and ABTS^●+^ assays and Trolox equivalents (TE, as µmol g^−1^) for each assay also. Data are shown as means ± standard deviations (SDs) for 3 samples of lipid extracts from minced salmon heads (*n* = 5).

DPPH^●^ Assay		ABTS^●+^ Assay	
Inhibition (%)	TE (µmol g^−1^)	Inhibition (%)	TE (µmol g^−1^)
10.76% ± 0.69	4.39 ± 0.28	15.41% ± 0.62	1.67 ± 0.06

## Data Availability

The data presented in this study are available on request from the corresponding author.

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
