# Peer review of "Unveiling the Lipid Features and Valorization Potential of Atlantic Salmon (Salmo salar) Heads"

_marinedrugs, 2024, doi:10.3390/md22110518_

Round 1
Reviewer 1 Report
Comments and Suggestions for Authors
The authors present a study of the lipid composition of minced salmon heads. The work is well intentioned and generally the data are well presented. i have a few suggestions for the authors below.
1. results section Figure 2. Is it possible to add a figure 2B with the total amount (quantified lipid moles/ gram of tissue) of each lipid class. This would be helpful and should be easy since the methodology for the lipidomics incorporates internal lipid standards.
2. similarly, can the internal standards and their peak areas be identified in the supplementary data. This would make the data provided potentially more useful to other readers.
3 lipids are susceptible to oxidation and can give rise to toxic oxidation products. I suggest the authors note this in their discussion,
see https://pubs.acs.org/doi/10.1021/acs.chemrev.0c00761.
Author Response
Reviewer #1
The authors present a study of the lipid composition of minced salmon heads. The work is well intentioned and generally the data are well presented. i have a few suggestions for the authors below.
- results section Figure 2. Is it possible to add a figure 2B with the total amount (quantified lipid moles/ gram of tissue) of each lipid class. This would be helpful and should be easy since the methodology for the lipidomics incorporates internal lipid standards.
Reply: Unfortunately, it is not possible to calculate the amounts/percentages of lipids per class in a manner that would be technically correct. For example, for the main class, triglycerides (TG) (and also diglycerides, DG), we were not able to use class specific standards. We do have the standards (isotopic standards) but we are not able to solubilize them adequately to use them in our MS runs. In the case of TGs and DGs the normalization of the peak areas was performed using PC 28:0 as standard (the standard used with closer structure and closer retention time to these classes). That is why we call this a “semi-quantification”:
“Analysis of the lipidome of the minced salmon heads by LC-MS allowed the identification and semi-quantification of 150 different lipid species (Fig. 2).” (beginning of the section 2.3. Lipidome characterization).
This is also why the realization that TGs were the most abundant class was based on the quantification of polar lipids (1.16% of total lipid, Table I) revealing that the content in neutral lipids was vastly higher, and the observation of the total ion count chromatograms (Supplementary Fig. 3) clearly highlighting TG peaks to be very clearly the largest. Anyway, the amounts of the polar lipids identified should be negligible.
Ultimately, it would not be correct to calculate absolute amounts without having used standards specific to each lipid class, and also without using calibration curves built with the standards. We are aware of the limitations of our approach, which are the result of technical constraints, and that is why we never apply the term “quantification” when presenting/discussing the lipidomics results and use the term “semi-quantification” instead. Therefore, on the one hand, we are aware of the limitations that a semi-quantitative approach presents, but it is a valid approach used on many occasions in lipidomics studies when specific lipid standards are not available or when it is not possible to include them. On the other hand, the conclusions and overall discussion should not be compromised by these limitations, given the vestigial character of all lipid classes other than TGs. Anyway, and to make this issue absolutely clear, we included a sentence at the end of the discussion section acknowledging that this is, in fact, a limitation of the study:
“The limitations of this work include the fact that the lipidomics approach involved a semi-quantitative procedure, and therefore it is not possible to calculate quantitatively the amounts of each lipid species and lipid classes per quantity of biomass of co-product.”
- similarly, can the internal standards and their peak areas be identified in the supplementary data. This would make the data provided potentially more useful to other readers.
Reply: The internal standards were included and identified in Supplementary Table 2 and their peak areas included.
3 lipids are susceptible to oxidation and can give rise to toxic oxidation products. I suggest the authors note this in their discussion,
see https://pubs.acs.org/doi/10.1021/acs.chemrev.0c00761.
Reply: This issue has been addressed at the end of the first paragraph of page 10 (in the Discussion section):
“It is important to notice, however, that eventual uses of these resources for food/feed should take into account that lipids, particularly those containing PUFA, may be readily degraded by lipid oxidation reactions with the production of secondary oxidation products contributing to flavor deterioration and the occurrence of off-flavors [90-92]. Moreover, some of these products may present toxicity risks [93-95]. Therefore, the development of new approaches to detect and quantify oxidized lipids in resources like these co-products is important, and studies like this, characterizing the lipid components at a molecular level represent a first step in that direction.”
Reviewer 2 Report
Comments and Suggestions for Authors
The manuscript describes primarily lipid analysis of Atlantic salmon heads and its antioxidant property. The experiments were designed well, results are interesting. I would like to recommend it for publication after minor revision.
Comments –
1The title does not align with the content; it would be more appropriate to center the title around lipid analysis.
2In the results section, the author presents the lipid percentage in two different formats: 57.76% (DW, page 3, second paragraph) and 23.97% (actual percentage) in Table 1. It would be better to use the same format for both the text and the table, the same applies to other parameters.
3Even though short-chain fatty acids were listed in TAG structure in Supplementary Table 1, no fatty acids C8:0, C10:0 and C12:0 are described in Figure 1. Need more discussion.
4In section 2.3 author described the semi-quantification of the lipids, besides the relative percentage no quantitative data were described in the manuscript except phospholipids (Table 1). Figure 2 provides the number of lipids identified; it would be more effective to display the relative percentage of these molecules within the total lipid content.
5The TG 60:13 assignment seems wrong, TG with two DHA and oleic acid should have TG C62:13. It needs more explanation how the author determined the fatty acid acyl chain such as TG 52:3 (palmitic acid, oleic acid and linoleic acid). There are several possible combinations for TG 52:3 i.e, C16:0/18:0/18:3, 16:1/18:0/18:2, 16:3/18:0/18:0 etc. The same applies to the assignment of TG in Supplementary Table 1.
Author Response
Reviewer #2
The manuscript describes primarily lipid analysis of Atlantic salmon heads and its antioxidant property. The experiments were designed well, results are interesting. I would like to recommend it for publication after minor revision.
Comments –
- The title does not align with the content; it would be more appropriate to center the title around lipid analysis.
Reply: The title has been changed to: “Unveiling the Nutritional Lipid Features and Valorization Potential of Atlantic Salmon (Salmo salar) Heads”.
- In the results section, the author presents the lipid percentage in two different formats: 57.76% (DW, page 3, second paragraph) and 23.97% (actual percentage) in Table 1. It would be better to use the same format for both the text and the table, the same applies to other parameters.
Reply: The results in section “2.1. Elemental composition and biochemical characterization” have been corrected and expressed collectively in wet weight (WW) in accordance to what is presented in Table 1.
- Even though short-chain fatty acids were listed in TAG structure in Supplementary Table 1, no fatty acids C8:0, C10:0 and C12:0 are described in Figure 1. Need more discussion.
Reply: An explanation for this was included in the Discussion section (page 8, beginning of paragraph 3):
“The analysis of the lipidome of the minced Atlantic salmon heads allowed detecting TGs containing small chain fatty acids (8-12 carbons), as well as other odd-chain fatty acids in their fragmentation profile (MS/MS) that were not detected by GC-MS. This fact is explained by a lower sensitivity of the GC-MS technique with regard to high resolution LC-MS, and by the fact that the TG species containing those fatty acids were vestigial with regard to the majority TG species.”
- In section 2.3 author described the semi-quantification of the lipids, besides the relative percentage no quantitative data were described in the manuscript except phospholipids (Table 1). Figure 2 provides the number of lipids identified; it would be more effective to display the relative percentage of these molecules within the total lipid content.
Reply: We used a semi-quantitative approach in this work, since we were not able to include standards specific for the triglyceride (TG) and diglyceride (DG) classes. We do have the standards (isotopic standards) but we are not able to solubilize them adequately to use them in our MS runs. In the case of TGs and DGs the normalization of the peak areas was performed using PC 28:0 as standard (the standard used with closer structure and closer retention time to these classes). Therefore, it would not be technically correct to calculate the percentages of the classes and to sum the normalized peak areas to do it, without having used class-specific standards and respective calibration curves. Nevertheless, it is clear from the polar lipid quantification (1.16% of total lipid, Table I) that neutral lipids are in vast majority, and it is clear from the observation of the total ion count chromatograms (Supplementary Fig. 3) that TG peaks are significantly largest than anything else in the MS runs. Therefore, on the one hand, we are aware of the limitations that a semi-quantitative approach presents, but it is a valid approach vastly used in lipidomics studies when class-specific lipid standards are not available or when it is not possible to include them. On the other hand, the conclusions and overall discussion should not be compromised by these limitations, given the vestigial character of all lipid classes other than TGs. Anyway, and to make this issue absolutely clear, we included a sentence at the end of the discussion section acknowledging that this is, in fact, a limitation of the study:
“The limitations of this work include the fact that the lipidomics approach involved a semi-quantitative procedure, and therefore it is not possible to calculate quantitatively the amounts of each lipid species and lipid classes per quantity of biomass of co-product.”
- The TG 60:13 assignment seems wrong, TG with two DHA and oleic acid should have TG C62:13. It needs more explanation how the author determined the fatty acid acyl chain such as TG 52:3 (palmitic acid, oleic acid and linoleic acid). There are several possible combinations for TG 52:3 i.e, C16:0/18:0/18:3, 16:1/18:0/18:2, 16:3/18:0/18:0 etc. The same applies to the assignment of TG in Supplementary Table 1.
Reply: The reviewer is right, that was a typo (page 5, last paragraph). TG 62:13 was the TG presenting the most unsaturations (as you may confirm in the Supplementary Table 1) and is in fact composed of an oleic acid and two DHA. The error was corrected. The assignment was made based on the analysis of the fragmentation (MS/MS spectra) of TGs (and of lipid species of the other classes for that matter) and the presence of the m/z signaling the presence of the fatty acids. In the case of TGs there are in fact many possibilities/combinations in terms of fatty acid composition. For example, for TG 48:3 we have found fragments indicating the possibility of 10 different viable fatty acids present by analyzing the MS/MS spectrum (12:0, 14:0, 14:1, 16:0, 16:1, 16:2, 18:1, 18:2, 18:3 and 20:3). Many others had 7, 8, 9 fatty acid possibilities for combinations. That creates an enormous amount of possible combinations and that left us with two options: present all the fatty acids indicated by the analysis of the MS/MS spectra or just indicate the most abundant possibility. We found that the later, including only the most abundant combination (combinations including the largest peaks in the MS/MS spectrum), would be more informative as some of the remaining fatty acid characteristic fragments were just visible. We explained that approach/option in the legend of the Supplementary Table 1: “c: only the most abundant TG species is presented.” We have included a more detailed explanation in the 4.7. Reverse phase liquid chromatography–mass spectrometry (C18–LC–MS) section to make the prosses of species assignment clearer:
“Lipostar annotations and assignments were manually confirmed based on the analysis of the fragmentation (MS/MS spectra) and the presence of the m/z signaling the presence of the fatty acids and class characteristic fragments.”
Reviewer 3 Report
Comments and Suggestions for Authors
Some suggestions were as follows:
1. The article lacked of the line numbers, and it caused difficults for the reviewers to point the defects of this paper.
2. What is the innovation of this article? This paper only used the heads of Atlantic salmon (Salmo salar) for lipid research, but what about the heads from other marine sources of fish heads? The author should compare and list of other sources in the article?
3. What about the content of enriched-DHA / EPA in the fatty acid chains of PC, PE, PI, and PS in the lipid that extracted from heads of Atlantic salmon (Salmo salar)? The authors needed to mention them in this paper.
4. Health benefits of dietary marine DHA/EPA-enriched glycerophospholipids needed mention in the Introduction, the authors can refer to https://doi.org/10.1016/j.intimp.2024.112895, https://doi.org/10.1016/j.plipres.2019.100997.
5. The defects or further studies of this study needed to mention in the Conclusions.
Author Response
Reviewer #3
Some suggestions were as follows:
- The article lacked of the line numbers, and it caused difficults for the reviewers to point the defects of this paper.
Reply: We apologize for the inconvenience. Amidst all the formalities involved in preparing a submission, it is always possible that something may be overlooked…
- What is the innovation of this article? This paper only used the heads of Atlantic salmon (Salmo salar) for lipid research, but what about the heads from other marine sources of fish heads? The author should compare and list of other sources in the article?
Reply: The innovation of the article was the full characterization of the lipidome of these resources for the first time, as we mentioned at the end of the Introduction section:
“Moreover, since available studies characterizing the lipid content in salmon co-products at a molecular level only focused on phospholipids, we aim to extend the characterization of minced salmon heads to its whole lipidome. “
In this case, this was a pertinent addition to was described before, since neutral lipids (and TGs in particular) are by far the most abundant lipid class in these resources, and they had not been thoroughly characterized yet. We believe that this characterization, highlighting good nutritional characteristics and a promising composition of TGs in particular in terms of their fatty acid content (omega-3 in particular), may raise interest in the utilization of these co-products. Moreover, this characterization may help direct these co-products for the most appropriate and profitable applications. This is the innovative aspect of this study, and we believe it is well discussed and explained in the manuscript.
Salmon heads are pertinent in this sense and very justifiably interesting for this type of characterization, given the importance of the species for human nutrition and the profuse production of these co-products worldwide. Many other species are commercialized whole (with heads), while others are just not that relevant or do not generate amounts of co-products globally that would justify the same type of interest. There are only a few studies characterizing the heads of other fish species, and we have now listed them in the introduction, as requested:
“Other studies using lipidomics tools to characterize the lipid content of the heads of other fish species include the silver carp (Hypophthalmichthys molitrix) [16], the Pacific blue mackerel (Scomber australasicus) [17] and King salmon (Oncorhynchus tshawytscha) [15], in the later species using a nuclear magnetic resonance (NMR), and all focusing specifically on the content in omega-3 containing phospholipids.”
Comparison of results is difficult because the available bibliography only focuses on the phospholipid content of fish heads and their composition in omega-3 in particular, and the content in polar lipid in our extracts is almost negligible. However, we did include a discussion about this issue in the Discussion section (page 9, paragraph 3):
“With regard to works characterizing the heads of other fish species by lipidomics tools, the available studies are difficult to compare to ours because they focused spe-cifically on the content in omega-3 containing phospholipids. A study characterizing silver carp heads phospholipids detected lipid species from the LPC, LPE, LPG, LPI, LPS, phosphatidic acid (PA), PC, PE, PG, PI and PS classes [16]. Here we were not able to detect lipids from the LPG, LPI, LPS, PA and PI class, probably because of the fact that polar lipid content is almost negligible in total lipid extracts of minced salmon heads. With regard to the classes that we were able to detect, this study reported LPC 16:0, LPE 22:6, PC 34:1, PE-O 34 :2, PG 34 :1 and PS 36 :1 to be the most abundant lipid species of their class [16]. In our study, although we were only able to detect a restrict-ed number of phospholipid species, we were able to detect LPC 16:0 and PC 34:1, and LPE 22:6 was in fact the only member of its class that we detected. Two other studies investigated the composition of phospholipids in omega-3 fatty acids in Pacific blue mackerel [17] and King salmon [15] heads, although using approaches based in NMR. Both studies reported remarkable contents in phospholipid with esterified omega-3 fatty acids especially in the heads of Pacific blue mackerel (40.9%). In our study, a deeper characterization of the phospholipid content was somewhat undermined by the overwhelmingly larger content of neutral lipids, but we do report PC 38:6 and PC 42:6 to be the most abundant lipid species of their class, while the only PEs (PE 34:5 and PE 44:12) and PG (PG 34:5) detected also contain omega-3 fatty acids, which seems to confirm the high prevalence of omega-3 fatty acids in the phospholipids of these re-sources.”
- What about the content of enriched-DHA / EPA in the fatty acid chains of PC, PE, PI, and PS in the lipid that extracted from heads of Atlantic salmon (Salmo salar)? The authors needed to mention them in this paper.
Reply: This question was in the previous reply (to question 2):
“Both studies reported remarkable contents in phospholipid with esterified omega-3 fatty acids especially in the heads of Pacific blue mackerel (40.9%). In our study, a deeper characterization of the phospholipid content was somewhat undermined by the overwhelmingly larger content of neutral lipids, but we do report PC 38:6 and PC 42:6 to be the most abundant lipid species of their class, while the only PEs (PE 34:5 and PE 44:12) and PG (PG 34:5) detected also contain omega-3 fatty acids, which seems to confirm the high prevalence of omega-3 fatty acids in the phospholipids of these re-sources.”
- Health benefits of dietary marine DHA/EPA-enriched glycerophospholipids needed mention in the Introduction, the authors can refer to https://doi.org/10.1016/j.intimp.2024.112895 https://doi.org/10.1016/j.plipres.2019.100997.
Reply: We are very aware of the health benefits proposed for marine phospholipids or phospholipids containing omega-3 fatty acids as we have mentioned them in previous publications. However, in the case of the resources we studied, the content in phospholipids is almost negligible with regard to the total lipid content (1.16±0.17%). Therefore, from a valorization or prospection perspective, we find the content in these phospholipids to be irrelevant for being so limited in this case. If we are to promote the use of these co-products for prospective new applications, it will not be possible to do it on the basis of a 1.16% content. In this case, the valorization will have to be based on the content of the major components, namely TGs. Nevertheless, we have included a small introduction/mention to this issue in the Introduction section (page 2, paragraph 2):
“Other studies using lipidomics tools to characterize the lipid content of the heads of other fish species include the silver carp (Hypophthalmichthys molitrix) [16], the Pacific blue mackerel (Scomber australasicus) [17] and King salmon (Oncorhynchus tshawytscha) [15], in the later species using a nuclear magnetic resonance (NMR), and all focusing specifically on the content in omega-3 containing phospholipids. This interest in the content in omega-3-containing phospholipids is justified by the wide range of health benefits ascribed to these compounds when acquired through the diet [18, 19].”
- The defects or further studies of this study needed to mention in the Conclusions.
Reply: The limitations of the work are now acknowledged in the manuscripit. We opted to include these considerations at the end of the Discussion section, because we believe it makes more sense as we are discussing the work, and also because we do not want to change the focus of the Conclusions section. Our perspectives about possible follow-up work were also included:
“The limitations of this work include the fact that the lipidomics approach involved a semi-quantitative procedure, and therefore it is not possible to calculate quantitatively the amounts of each lipid species and lipid classes per quantity of biomass of co-product. Moreover, since the objective of this work was to perform a characterization of the lipid content as thoroughly as possible, we used the standard methods (Bligh and Dyer) known to assure an optimized extraction of total lipid and lipid class coverage. However, these classical methods use solvents that are not compatible with uses in the food/feed industry and, therefore, the next steps of the valorization of these co-products include testing the use of green methods/solvents for the lipid extraction in these resources. Another limitation was the fact that we were only able to bioprospect biological activity in lipid extracts from these resources in terms of antioxidant activity. In the future, it will be important to extend this screening to other possible manifestations of biological activity in order to improve the possibilities of unveiling other innovative uses for these resources in possible higher-end applications. Finally, future work also includes the planning and testing of biorefinery setups able to optimally recover and guarantee the quality of both lipid and protein content from these co-products.”
Round 2
Reviewer 3 Report
Comments and Suggestions for Authors
The authors revised their paper and it can be accepted in the present form.